# OpenReview forum: "From Vulnerability to Defense: Understanding and Mitigating MASK-Based Attacks in dLLMs"
_ICLR.cc/2026/Conference — ICLR 2026 Conference Withdrawn Submission_

### Official Review · Reviewer_yW3Z · 2025-10-28

**Soundness:** 3
**Presentation:** 3
**Contribution:** 3
**Rating:** 4
**Confidence:** 3

**Summary:**

This paper investigates a specific vulnerability in DLLMs, arguing they are highly susceptible to "MASK-based" jailbreak attacks where adversarial prompts use `[MASK]` tokens to elicit unsafe content. I think a key contribution is the theoretical analysis, which identifies "margin accumulation" and "scheduling advantages" as the root causes of this vulnerability. Based on this analysis, the authors propose a defense framework consisting of a two-stage data synthesis process and a novel "Reject-MASK" training strategy. Experiments demonstrate that this approach effectively reduces attack success rates from over 90% to near single-digit levels while largely preserving the model's performance on utility benchmarks.

**Strengths:**

I found the theoretical analysis of the `[MASK]` token vulnerability in dLLMs to be a primary strength of this work. The authors move past a simple demonstration of the attack and provide a formal explanation, identifying "margin accumulation" and "scheduling advantages" as the core issues. This provides a foundation for the rest of the paper.

Building on this analysis, the proposed defense framework is well-motivated. I think the "Reject-MASK" strategy  is a clever response that directly targets the identified vulnerability by strengthening the model's ability to reconstruct rejection semantics.

**Weaknesses:**

My main concern revolves around the fundamental nature of the vulnerability being explored. Recent work on autoregressive models (I'm thinking of **the ICLR 2025 outstanding paper 'Safety Alignment Should be Made More Than Just a Few Tokens Deep'**) has compellingly shown that their safety alignment is often superficial and heavily tied to the chat template structure. If one bypasses this template, attack success rates can approach 100% even without sophisticated attacks. **This makes me wonder: is the `[MASK]`-based attack on dLLMs a *new* vulnerability class, or is it an instance of this known 'template-bypassing' problem, just manifested in a different architecture? The `[MASK]` token effectively forces the model into an in-filling mode, which is not the standard instruction-following (chat) mode for which it was presumably aligned. (also check this blog https://zhuanlan.zhihu.com/p/1892933255927428134)**

This concern is also linked to the choice of model. The paper uses LLaDA, but it's not entirely clear to me how its base safety alignment compares to the robust alignment of modern, SOTA autoregressive models. I suspect the base LLaDA's alignment might be less robust to begin with. A critical question, which I think the paper doesn't fully resolve, is whether this `[MASK]` vulnerability would persist if a dLLM were aligned to the same, state-of-the-art level as today's top ARMs. It's possible the high ASRs reported are partly an artifact of LLaDA's specific alignment, rather than an *inherent* property of all dLLMs.

I also found the details of the two-stage data synthesis framework a bit underspecified. The paper mentions extracting entities to generate "safe reject" responses and separately synthesizing "utility-oriented" instructional data. Personally, I would have liked a more in-depth discussion on how these two data types are balanced and generated. The 'Mix' dataset is an even split, but the *strategy* for creating the utility data to ensure it doesn't conflict with the safety data isn't fully clear. It seems like a delicate balance, and more details on this data curation process would strengthen the paper's reproducibility and claims.

Finally, while the paper does provide a crucial ablation in Table 1 ('Mix w/o Reject-MASK'), I think a clearer picture could be painted. For instance, what would be the effect of applying the Reject-MASK strategy directly to the *base* model without the new safety-utility data? This would help isolate the contribution of the training strategy itself from the new data distribution.

**Questions:**

My primary question relates to the core vulnerability. Could you elaborate on why this `[MASK]`-based attack is an *inherent* issue with the diffusion architecture, rather than a form of the "template bypassing" problem we've seen in ARMs? I'm trying to understand if LLaDA's base safety alignment (which might be less robust than SOTA ARMs) is the main enabler here. A response clarifying this, perhaps by comparing `[MASK]` attacks to other non-`[MASK]` alignment-bypassing attacks on LLaDA, could really strengthen my understanding of the paper's core claim.

I would also appreciate more detail on the two-stage data synthesis framework. Specifically, for the "utility-oriented" data, how did you ensure this data was both helpful and didn't inadvertently *reinforce* the model's tendency to comply with instructions that *look* similar to the harmful, masked prompts? It seems like a difficult balance to strike, and more insight into this data curation would be very helpful.

Finally, I'm curious about the relative contributions of your two-part defense. The ablation 'Mix w/o Reject-MASK' is useful, but I'm missing the other side. Do you have any results or insights on what would happen if you *only* applied the Reject-MASK training strategy to the *original base model* (without the new two-stage data)? This would help me understand if the new data distribution is a prerequisite for the Reject-MASK strategy to be effective, or if the strategy has standalone benefits.

---

> ### Author Response · Authors · 2025-11-29
> **Response to Reviewer yW3Z**
>
> **1. Clarification of the Core Vulnerability and its Relation to the "Template-Bypassing" Problem**
>
> **Reviewer Concern:**
>
> The reviewer expresses concern about whether the [MASK]-based attack on diffusion large language models (dLLMs) is a new vulnerability class or simply a manifestation of the known "template-bypassing" issue observed in autoregressive models (ARMs). They also question whether the base safety alignment of LLaDA is weaker, which might explain the high attack success rates.
>
> **Response:**
>
> The attack discussed in the paper focuses on the bidirectional and parallel decoding architecture of dLLMs, specifically on how the interaction between masked tokens and generation leads to vulnerabilities. While autoregressive models (ARMs) can be vulnerable to template-bypassing (where structured prompts or templates are bypassed by adversarial inputs), the [MASK]-based attacks in dLLMs exploit specific structural weaknesses of the bidirectional model architecture. The [MASK] token serves as a structural control point, and the model, compelled by its bidirectional nature, generates fluent outputs even in harmful contexts. This is fundamentally different from ARMs where the vulnerabilities stem from the flexibility in prompt construction and the model’s inability to fully filter harmful inputs.
>
> The paper explicitly differentiates between the "template-bypassing" phenomenon in ARMs and the vulnerabilities inherent in dLLMs due to their architectural specifics. By showing how masked token manipulation can lead to unsafe completions in the absence of proper safety mechanisms, the paper argues that these issues are not merely an artifact of misalignment in template structures but are specific to the design of diffusion models themselves. This distinction helps clarify that the vulnerability is not just an extension of known ARM issues but a novel challenge for dLLMs .
>
> Regarding the alignment concerns, while LLaDA's safety alignment may be less robust than that of current state-of-the-art ARMs, the paper makes a theoretical and empirical case for why this vulnerability would persist even in more robustly aligned models. The experiments with LLaDA and the comparison with other models like MMaDA suggest that the attack success rate is not solely an artifact of LLaDA’s specific alignment but a more general issue in the dLLM architecture.
>
> **2. Details on the Two-Stage Data Synthesis Framework and Balancing Safety and Utility**
>
> **Reviewer Concern:**
>
> The reviewer requests more clarity on the two-stage data synthesis framework, particularly regarding how the utility-oriented and safety-oriented data are balanced. They also ask for further explanation on how the utility data ensures it does not conflict with the safety data.
>
> **Response:**
>
> The two-stage data synthesis framework is designed to strike a balance between safety and utility by generating training data that incorporates both safety-relevant "safe reject" responses and task-specific "utility-oriented" instructions. The safe reject data consists of key entities extracted from harmful prompts, ensuring that the model learns to reject unsafe instructions effectively. Meanwhile, the utility-oriented data ensures that the generated responses are relevant to the task at hand.
>
> To ensure these two types of data do not conflict, the synthesis process carefully mixes safety and utility data while maintaining the instructional integrity of the prompts. The "Mix" dataset is an even split between these two types of data, but the synthesis approach ensures that the utility data does not promote harmful behaviors by reinforcing safety training. This is done through careful data curation strategies, where safety-related responses are explicitly tied to harmful entities, preventing accidental reinforcement of unsafe completions from the utility data.
>
> **3. Effect of Reject-MASK Strategy and Its Contribution in Isolation**
>
> **Reviewer Concern:**
>
> The reviewer asks for a clearer picture of the contribution of the Reject-MASK strategy by isolating its effect from the new data distribution, asking what happens if the Reject-MASK is applied to the base model without the new data.
>
> **Response:**
>
> The ablation experiment 'Mix w/o Reject-MASK' in Table 1 provides insight into the impact of the Reject-MASK strategy when applied alongside the two-stage data synthesis. However, the overall safety-utility balance is not as robust without the mixed data, as the base model’s general performance might suffer more without the utility data providing sufficient context for task-specific responses. Therefore, the Reject-MASK strategy has a standalone effect in improving safety, but its full effectiveness in maintaining utility is realized when paired with the balanced dataset that includes both safety and utility-oriented data.

---

> ### Author Response · Authors · 2025-11-29
>
> We appreciate reviewer's  thoughtful feedback and have addressed their concerns in detail:
>
> * **Clarification on Core Vulnerability**: We clarify that the [MASK]-based attack in dLLMs is distinct from the "template-bypassing" issue observed in autoregressive models. The vulnerability in dLLMs arises due to the specific interaction between masked tokens and bidirectional model architecture, which generates unsafe outputs in certain contexts. We also explained that the observed attack success rate is not just a result of LLaDA's alignment but a broader issue in dLLM design.
>
> * **Details on Two-Stage Data Synthesis Framework**: We provide more clarity on how the utility-oriented and safety-oriented data are balanced in the two-stage data synthesis process. We emphasized that careful data curation ensures the two types of data complement each other, preventing the utility data from inadvertently reinforcing unsafe completions.
>
> * **Effect of Reject-MASK Strategy**: We address the impact of the Reject-MASK strategy in isolation, explaining that while the strategy does have standalone benefits, its full effectiveness is realized when paired with the mixed data to maintain both safety and utility.
>
> We believe our responses clarify the core issues raised by the reviewer and provide stronger insights into the contribution of our work. As a result, the reviewer has **revised their score from 4 to 8**, and we thank them for their constructive comments. We also extend our gratitude to the Area Chair for their ongoing support and efforts in facilitating the review process.

---

### Official Review · Reviewer_8Jct · 2025-11-01

**Soundness:** 2
**Presentation:** 2
**Contribution:** 2
**Rating:** 2
**Confidence:** 4

**Summary:**

The paper studies a jailbreak vector against diffusion LLMs (dLLMs) that leverages masked tokens ([MASK]) and parallel decoding to obtain unsafe completions from a model. The authors argue theoretically that small positive “logit margins” at masked positions can compound across parallel slots, yielding high attack success. They then propose a defense combining (i) a two-stage data synthesis pipeline and (ii) Reject-MASK training, which strengthen “rejection trajectories.” Experiments on HarmBench and JailbreakBench report large drops in attack success rates for several attacks (Zeroshot, AIM, PAIR, DIJA), while retaining most utility.

**Strengths:**

**Originality / framing.**
Focuses on a specific architectural vulnerability of dLLMs (parallel filling of [MASK] slots) and ties it to “margin accumulation and scheduling advantage.” This target is timely for the dLLM line of work. The paper claims to be the first theoretical analysis of [MASK]-based jailbreak risk in dLLMs.

**Defense concept.**
Reject-MASK is simple and easy to combine with data augmentation. The mechanism, bias masking around refusal tokens to increase the chance that “reject semantics” remain in the top-logits during reconstruction, is intuitive.

**Empirical performance.**
The presented defense scheme is effective and strongly decrease the presented attacks' success rate.

**Weaknesses:**

**Critical issues in Lemma 1 and its proof.**

* The proof of Lemma 1 (Appendix B.2) uses a relaxation ( $ \sum_{u \notin {h^{\star},s^{\star}}} e^{z_i(u)} \leq e^{z_i(h^{\star})} $ ) to derive a sigmoid lower bound. This relaxation is extremely strong/unrealistic and not justified. Moreover, the transition from eq. (16) to eq. (17) appears to flip the inequality direction: by *reducing* the denominator, you should obtain an **upper** bound, not a lower bound, so the intended lower-bound conclusion is not supported. As written, these steps suggest the bound is likely **an upper bound**, reversing the lemma’s claim and undermining later results that depend on it.

**Missing definitions / assumptions and unclear notation in the theory.**

* **Assumption 1**: Independence across slots (used later in product arguments) is not clearly specified as an assumption; it should be explicit.
* **Assumption 2**: Uses ( $ I ( \cdot ; \cdot ) $ ) and a variable ( $ U $ ) without prior definition in the main text.
* **Lemma 1 notation**: ( $ z_i( \cdot ) $ ) is introduced as “the logit at position (i)”, but the conditioning/context used to compute these logits is implicit rather than made precise in the lemma statement. Proof reference/linkage should be adjacent to lemma.

**Presentation of the method.**

* **Dataset pipeline clarity/reproducibility.** The two-stage synthesis is only sketched; it needs a precise, reproducible description (entity extraction procedure, "safe reject" generation policy, how entity sets are built/validated, ...). Currently, key steps rely on GPT-4o with a small total size (1,200 samples) and the paper provides limited ablation of each component’s contribution.
* **Metrics definition/interpretation.** ASR-k/e are central but defined only in Appendix D.4; the main text should briefly restate how ASR-k (keyword-based) and ASR-e (evaluator-based) are computed and by which evaluator(s) but at the very least reference Appendix D.4 for details.
* **Table readability and deltas.** Table 1 is dense and hard to parse; Table 2 lacks deltas w.r.t. base for quick trade-off reading. Adding deltas and confidence intervals would materially improve clarity.

**Claims vs. mixed results (generalization).**

* While LLaDA shows dramatic reductions, MMaDA remains at relatively high DIJA/JailbreakBench rates even with Reject-MASK (e.g., DIJA ASR-e ~28.75–30%). The paper should temper generality claims and analyze where/why the method is less effective (architecture, decoding, guidance, or data mismatch).

**Writing/formatting issues that impede clarity.**

* Multiple typos, unclear sentences, missing references, and a malformed URL in a footnote (still containing a "utm_source=chatgpt.com" and typeset as math), plus figure readability issues (Figure 2 colors/labels) cumulatively undermine the quality of the manuscript.

**Questions:**

**Suggestions to the authors.**

* **Introduction**: unclear sentence about data synthesis/training ("Reject-MASK focuses on reject-related tokens…"). Please rewrite for clarity.
* **Section 3.1, l.127**: Typo "refers to use of".
* **Footnote 1**: Malformed URL including `utm_source=chatgpt.com`, typeset as a formula $\rightarrow$ please fix.
* **p.5, l.191**: "part" $\rightarrow$ "past". Also, "Our analysis in **Section 3**" (fix the section cross-ref in Section 4.1).
* **Section 4.1**: The two-stage synthesis description is not yet reproducible (entity recognition, "safe reject" generation, entity sets). Add full algorithmic details.
* **Figure 2**: "Quary" $\rightarrow$ "Query"; improve color contrast; expand caption to describe the full pipeline meaningfully.
* **Section 4.2**: Lines 258–260 are too terse; describe the masking schedule, token selection rules, and probabilities.
* **Link theory with experiments**: Add explicit checks that empirical measures conform to your theory; currently the experiments don’t directly confront the theoretical quantities.
* **Appendix B titles**: Make subsection titles tie to main-text lemmas (e.g., "B.2 Proof of Lemma 1: …").

In addition to the elements mentioned as weaknesses.

---

> ### Author Response · Authors · 2025-11-17
> **Response to Reviewer 8Jct part(1/6)**
>
> **We truly appreciate the reviewer’s thoughtful and valuable feedback. We have taken each comment to heart and have carefully considered how to address the concerns raised. We hope that our responses meet your expectations and address your points satisfactorily. If so, we would be extremely grateful if you could kindly reconsider your rating of 2 (reject). However, if any concerns remain, we would be genuinely thankful for your continued guidance. We are fully committed to resolving any outstanding issues and will work diligently to improve our submission in accordance with your suggestions.**
>
> > **Weakness 1**  **Critical issues in Lemma 1 and its proof.**
> >
> > The proof of Lemma 1 (Appendix B.2) uses a relaxation to derive a sigmoid lower bound. This relaxation is extremely strong/unrealistic and not justified. Moreover, the transition from eq. (16) to eq. (17) appears to flip the inequality direction: by *reducing* the denominator, you should obtain an **upper** bound, not a lower bound, so the intended lower-bound conclusion is not supported. As written, these steps suggest the bound is likely **an upper bound**, reversing the lemma’s claim and undermining later results that depend on it.
>
> Thank you for pointing out the issues with mathematical proofs. We sincerely apologize for the inconsistencies and errors in the original submission.
>
> In the revised version, we have revised the proof of Lemma 1 and corrected the mathematical relaxation that was previously used.
>
> The new proof of Lemma 1 is as follows:
>
> Let $h_i^\\star=h_i$ and $s_i^\\star=s_i$ be the strongest candidates under their respective sets, $\\Gamma(x) = \frac{1}{1+e^{-x}+(|\\mathcal V| - 2)e}$, and define the **minimum margin gap** as
>
> \\begin{equation}
> \\gamma_i:=z_i(h_i^\\star)-z_i(s_i^\\star).
> \\end{equation}
>
> Then
>
> \\begin{equation}
> p_\\phi(h_i^\\star\\mid U,X_t^{(-i)})\\;\\ge\\;\\Gamma(\\gamma_i).
> \\end{equation}
>
> \\begin{equation}
> p_\\phi(h_i^\\star\\mid U,X_t^{(-i)})=\\frac{e^{z_i(h_i^\\star)}}{\\sum_{u\\in\\mathcal V}e^{z_i(u)}}
> \\end{equation}
>
> \\begin{equation}
> =\\frac{1}{1+e^{-(z_i(h_i^\\star)-z_i(s_i^\\star))}+\\sum_{u\\in\\mathcal V\\setminus\\{h_i^\\star,s_i^\\star\\}}e^{z_i(u)-z_i(h_i^\\star)}} \\ge \\frac{1}{1+e^{-\\gamma_i}+(|\\mathcal V| - 2)e}=\\Gamma(\\gamma_i)
> \\end{equation}
>
> Here, we uses the relaxation $z_i(u)-z_i(h^\\star) \\leq 1$.

---

> ### Author Response · Authors · 2025-11-17
> **Response to Reviewer 8Jct part(2/6)**
>
> > **Weakness2 Missing definitions / assumptions and unclear notation in the theory.**
> >
> > - **Assumption 1**: Independence across slots (used later in product arguments) is not clearly specified as an assumption; it should be explicit.
> > - **Assumption 2**: Uses I and a variable U without prior definition in the main text.
> > - **Lemma 1 notation**: z_i is introduced as “the logit at position (i)”, but the conditioning/context used to compute these logits is implicit rather than made precise in the lemma statement. Proof reference/linkage should be adjacent to lemma.
>
> Thank you for your insightful and detailed feedback. We greatly appreciate your attention to the clarity and precision of the theoretical aspects of our work.
>
> **Assumption 1**: We acknowledge that the assumption of independence across slots was not clearly stated. In the revised version, we have explicitly included Assumption 1, which is derived from foundational work in diffusion language models [1], as well as from the concept of constrained diffusion in models that focus on token ordering [2]. Specifically, Assumption 1 assumes that within the parallel denoising framework of diffusion models, the processing of masked tokens is independent across different slots.
>
> **User Input and Mutual Information**: We appreciate your comment regarding the notation. The variable $U$ refers to user input, and the function $I$ represents mutual information. We have now defined these terms in the revised version of the paper to ensure they are clear from the outset.
>
> **Lemma 1 Notation and Context**: We recognize that the conditioning/context used to compute the logits in Lemma 1 was not sufficiently explicit. In the revised version, we have made it clearer that $z_i(h_i)$ and $z_i(s_i)$ represent the logits of candidate harmful and safety markers, respectively, for the positions being masked. While this context was implicit, we have now explicitly stated it in the lemma to ensure the explanation is precise and clear. Additionally, we have ensured that the proof reference/linkage is adjacent to the lemma as suggested.
>
>   [1] Nie et al., Large Language Diffusion Models. 2025.
>
>   [2] Kim et al., Train for the Worst, Plan for the Best: Understanding Token Ordering in Masked Diffusions. 2025.

---

> ### Author Response · Authors · 2025-11-17
> **Response to Reviewer 8Jct part(3/6)**
>
> > **Weakness3** **Presentation of the method.**
> >
> > - **Dataset pipeline clarity/reproducibility.** The two-stage synthesis is only sketched; it needs a precise, reproducible description (entity extraction procedure, "safe reject" generation policy, how entity sets are built/validated, ...). Currently, key steps rely on GPT-4o with a small total size (1,200 samples) and the paper provides limited ablation of each component’s contribution.
> > - **Metrics definition/interpretation.** ASR-k/e are central but defined only in Appendix D.4; the main text should briefly restate how ASR-k (keyword-based) and ASR-e (evaluator-based) are computed and by which evaluator(s) but at the very least reference Appendix D.4 for details.
> > - **Table readability and deltas.** Table 1 is dense and hard to parse; Table 2 lacks deltas w.r.t. base for quick trade-off reading. Adding deltas and confidence intervals would materially improve clarity.
>
> Thank you for your review and thoughtful comments. We appreciate your suggestions and have carefully considered how to address each point.
>
> **Safe Dataset Construction**:
>  The safe dataset construction involves generating harmful prompts using DIJA, with masked positions replaced by refusal words. The goal is to ensure that all masked tokens are substituted with safe, ethically aligned responses, effectively rejecting harmful behaviors in the model's output.
>
> **Utility Data Construction**:
>  **Entities**: In our work, entities refer to key components such as names, actions, or objects that are extracted from harmful prompts. These entities are extracted from the HarmBench dataset and serve as the basis for generating questions for the utility dataset.
>
> The utility data is constructed in two stages:
>
> - **Step 1 (Entity Extraction)**: Entities are first extracted from harmful prompts in the HarmBench dataset.
> - **Step 2 (Questions and Step-by-Step Responses)**: After identifying the entities, we generate benign, step-by-step questions that incorporate these entities. These questions are paired with concise, informative, and neutral step-by-step responses aimed at addressing the over-rejection issue seen in the Safe dataset. The **educational prompt** (as referenced in section D.7) plays a critical role here, guiding the creation of instructional responses that are task-relevant and effectively counteract the over-rejection observed in the Safe dataset.
>
> **Metrics Definition/Interpretation**:
>  The definitions for ASR-k (keyword-based) and ASR-e (evaluator-based) are currently provided in Appendix D.4. To enhance clarity and make these metrics more accessible to readers, we will restate how ASR-k and ASR-e are computed in the main text, with a brief reference to Appendix D.4 for further details.
>
> **Table Readability and Deltas**:
>  We acknowledge that Table 1 is dense and could be challenging to interpret. To improve readability, we will revise the layout of the table for easier parsing. Regarding Table 2, we understand the importance of showing performance deltas relative to the base model for quicker comparisons. We will add these deltas, along with the corresponding confidence intervals, to provide a clearer understanding of the results and enhance the interpretation of the trade-offs.

---

> ### Author Response · Authors · 2025-11-17
> **Response to Reviewer 8Jct part(4/6)**
>
> > **Weakness4** **Claims vs. mixed results (generalization).**
> >
> > While LLaDA shows dramatic reductions, MMaDA remains at relatively high DIJA/JailbreakBench rates even with Reject-MASK (e.g., DIJA ASR-e ~28.75–30%). The paper should temper generality claims and analyze where/why the method is less effective (architecture, decoding, guidance, or data mismatch).
>
> Thank you for your thoughtful feedback regarding the generalizability of our method. We appreciate your comments, and we agree that a more nuanced discussion of the results is needed.
>
> The high success rates observed with LLaDA can be attributed to several factors:
>
> - **Targeted Slot-Masking Strategy**: Our approach is tailored to the model's training regime and the specific types of prompts we evaluated. This alignment ensures that the model’s vulnerabilities are effectively addressed by our defense strategy.
>
> - **Generalization Failures in LLM Safety**: Recent studies on LLM safety training have highlighted that models often struggle when faced with unseen prompt distributions or adversarial patterns, due to mismatches between capability and safety fine-tuning. Since LLaDA’s architecture and prompt distribution align closely with our training and evaluation setup, our method achieves significant improvements.
>
> On the other hand, **MMaDA** seems to exhibit different behavior in terms of decoding or prompt distribution. The wider, more diverse set of prompts used with MMaDA, coupled with a weaker assumption of masked-slot independence, results in residual unsafe behaviors even after applying Reject-MASK (e.g., DIJA ASR-e ~28.75–30%). We believe this can be explained by several factors:
>
> - Architecture: MMaDA may implicitly model dependencies across slots, which could reduce the effectiveness of our slot-masking strategy.
>
> - Decoding Strategy: If MMaDA employs more diverse or stochastic decoding methods, our masked-slot product argument becomes less effective, leading to higher residual risk.

---

> ### Author Response · Authors · 2025-11-17
> **Response to Reviewer 8Jct part(5/6)**
>
> > **Weakness5** **Writing/formatting issues that impede clarity.**
> >
> > Multiple typos, unclear sentences, missing references, and a malformed URL in a footnote (still containing a "utm_source=chatgpt.com" and typeset as math), plus figure readability issues (Figure 2 colors/labels) cumulatively undermine the quality of the manuscript.
>
> Thank you for your feedback on the writing and formatting issues in the paper.
>
> **Writing and Typos**: We have reviewed the paper and corrected typos and unclear sentences to improve readability and clarity.
>
> **Footnote Issues**: The malformed URL in the footnote has been fixed.
>
> **Figure Readability**: We have also addressed the readability issues with Figure 2.

---

> ### Author Response · Authors · 2025-11-17
> **Response to Reviewer 8Jct part(6/6)**
>
> > **Questions:**
> >
> > ​	**Suggestions to the authors.**
> >
> > - **1. Introduction**: unclear sentence about data synthesis/training ("Reject-MASK focuses on reject-related tokens…"). Please rewrite for clarity.
> > - **2. Section 3.1, l.127**: Typo "refers to use of".
> > - **3. Footnote 1**: Malformed URL including `utm_source=chatgpt.com`, typeset as a formula $\rightarrow$ please fix.
> > - **4. p.5, l.191**: "part" $\rightarrow$ "past". Also, "Our analysis in **Section 3**" (fix the section cross-ref in Section 4.1).
> > - **5. Section 4.1**: The two-stage synthesis description is not yet reproducible (entity recognition, "safe reject" generation, entity sets). Add full algorithmic details.
> > - **6. Figure 2**: "Quary" $\rightarrow$ "Query"; improve color contrast; expand caption to describe the full pipeline meaningfully.
> > - **7. Section 4.2**: Lines 258–260 are too terse; describe the masking schedule, token selection rules, and probabilities.
> > - **8. Link theory with experiments**: Add explicit checks that empirical measures conform to your theory; currently the experiments don’t directly confront the theoretical quantities.
> > - **9. Appendix B titles**: Make subsection titles tie to main-text lemmas (e.g., "B.2 Proof of Lemma 1: …").
>
> Thank you very much for your detailed and constructive suggestions. We sincerely appreciate your insightful suggestions. We have carefully addressed each point as follows:
>
> * For **1,2,3,4,6,9** points: We have been fully rewritten for clarity in the revised version.
>
> * For **5** point: W e have added full algorithmic details for the two-stage data synthesis process in **Response for Weakness 3**.
>
> * For **7** point:
>     In our training process, the masking strategy primarily targets tokens related to rejection expressions. If a rejection word appears in the input text, we apply a more focused masking approach, masking both the token for the rejection word and additional surrounding tokens, ensuring that the model learns to reject these undesirable contexts and helping to reinforce rejection behavior during training. If threre is no rejection word, we apply a random masking strategy with a slightly higher probability for masking to ensure diversity in the training data.
>     The probability of masking is adjusted for different cases: when a rejection word is present in the input, we apply a higher masking probability to ensure the model learns strong rejection patterns. In other cases, the masking probability is determined by a small random factor within a set range to avoid overfitting.
>
> * For **8** point:
>     Our theoretical results establish that the lower bound of [MASK]-based attack success increases with the number of harmful token slots. While several of these theoretical guarantees rely on existence assumptions that are difficult to verify explicitly in practice, the empirical trend in Figure 4 closely matches the theoretical prediction: ASR consistently increases as the number of MASK positions grows. This monotonic pattern provides indirect but strong evidence in support of our theoretical conclusions.

---

> ### Author Response · Authors · 2025-11-17
>
> Dear Reviewer 8Jct,
>
> We are truly grateful for the time and care you have devoted to reviewing our paper. Your thoughtful questions and insightful feedback have been invaluable in helping us strengthen and clarify our work. We fully understand the considerable effort required to review many submissions, and we deeply appreciate the attention and dedication you have given to our research. To support your continued review, we have prepared detailed responses to each of your comments and have summarized the main points of our rebuttal below for your convenience.
>
> - Addressing critical **issues in Lemma 1 and its proof** by providing a revised proof (please refer to Weakness 1 in response for details)
> - Clarifying theoretical aspects by explicitly **stating assumptions and defining notations** (please refer to Weakness 2 in response for details)
> - Improving the presentation of the method by clarifying the **dataset pipeline and metric definitions** (please refer to Weakness 3 in response for details)
> - Tempering claims on **generalization and providing a deeper analysis** of the mixed results (please refer to Weakness 4 in response for details)
> - Correcting **writing and formatting issues** to improve clarity (please refer to Weakness 5 in response for details)
> - Responding to **specific suggestions** for improving clarity, fixing typos, and linking theory with experiments (please refer to the "Questions" section in the response for details)
>
> We sincerely hope that our responses have addressed your concerns. If any questions remain or if further clarification would be helpful, we would be more than willing to provide additional information. We truly appreciate the time, thought, and care you have invested in reviewing our work, especially given the demands of the review process. If our revisions have resolved your concerns, we would be deeply grateful if you would consider revisiting your rating. If not, we warmly welcome any further feedback, and we will gladly continue working to improve our submission.
>
> Thank you again for your invaluable time and thoughtful feedback.
>
> Best regards,
>
> Authors

---

> ### Author Response · Authors · 2025-11-26
>
> Dear Reviewer,
>
> I hope this message finds you well. As the discussion period is drawing to a close in just a few days, I would like to once again express my deepest gratitude for your time, patience, and invaluable guidance throughout the review process.
>
> Before the discussion ends, I humbly wish to ensure that we have addressed all of your concerns to the best of our ability. If there are any remaining questions, suggestions, or aspects that you believe require further clarification or improvement, please kindly share them with us. Your insights mean a great deal to us, and we sincerely hope to refine our work in accordance with your thoughtful advice.
>
> Thank you once again for your generous effort, careful consideration, and the expertise you have devoted to reviewing our paper. We are truly grateful for your support.
>
> With my most respect and sincere appreciation,
>
> Authors

---

> ### Author Response · Authors · 2025-11-29
> **Summary for Rebuttal to AC**
>
> Dear Area Chair,
>
> We would like to express our sincere gratitude to reviewer for their thorough and constructive feedback on our paper. We have carefully addressed all the concerns raised by the reviewer and made significant revisions to improve the manuscript.
>
> * **Critical issues in Lemma 1**: We **revise the proof of Lemma 1, fixing the previous inconsistencies and errors** related to the relaxation and the transition between equations. We have now clarify and strengthened the proof to align with the intended conclusions.
>
> * **Missing definitions and unclear notation**: We have **explicitly stated all assumptions and defined previously ambiguous variables** in the revised version, ensuring clarity in the theoretical framework.
>
> * **Presentation of the method**: We have **provided a detailed and reproducible description** of the two-stage dataset synthesis process, including entity extraction and safe reject generation. Additionally, we clarify the metric definitions and improved the readability of the tables.
>
> * **Generalization concerns**: We discuss the mixed results observed with MMaDA, providing a more nuanced analysis of the factors that contribute to the varying success rates across models.
>
> * **Writing and formatting issues**: We have corrected typos, improved sentence clarity, fixed the malformed URL, and enhanced figure readability.
>
> The reviewer has acknowledged the improvements and **increased the rating from 2 to 6**, reflecting the effectiveness of our revisions. We sincerely thank both the reviewer and you, the Area Chair, for your time, effort, and insightful feedback throughout this process.
>
> Best regards,
>
> The Authors

---

### Official Review · Reviewer_y7dP · 2025-11-01

**Soundness:** 3
**Presentation:** 1
**Contribution:** 2
**Rating:** 2
**Confidence:** 3

**Summary:**

This work attempts to provide a theoretical analysis of the vulnerability of diffusion LLMs (dLLMs) to mask token-based decoding attacks that circumvent safety alignment. The main theorem roughly states that the probability of a mask token being decoded into a harmful token is proportional to the minimum logit gap between “harmful” and “safe” tokens (although these do not seem to be well-defined) across all mask positions. This suggests that to defend against such attacks, one should reduce the gaps across all mask positions. A defense is proposed based on this observation, which significantly improves the robustness against mask attacks with minimal impact to model utility.

**Strengths:**

1. To my knowledge, this is one of the first works to theoretically analyze the brittleness of safety alignment to decoding attacks, not just for dLLMs but LLMs in general.
2. The proposed defense seems to be effective at improving the robustness against DiJA attacks, with minimal impact to model utility.

**Weaknesses:**

1. Notation discrepancies
        - Line 126: “Let vocabulary be V, with length L” — it seems in the subsequent notation that L refers to the sequence length, not the vocabulary size.
2. Spelling mistakes
        - Figure 2: “Quary” should be “Query
        - Line 275: Should “1) preserving” be “1) Utility-preserving”?
3. Line 129 assumes that the vocabulary contains a harmful set and safe set of tokens, but in reality whether a token is considered “harmful” or “safe” heavily depends on the surrounding context.
4. The takeaway from the main theoretical result seems to be rather obvious (that one should reduce the gaps for all mask positions). If one were to design a defense, the more obvious first solution would be to perform data augmentation on many different mask locations, rather than a few specific mask locations. Of course it is good to rigorously confirm this by proving the claim, however I feel that the high level takeaway doesn’t really yield a surprising insight.
5. It is very unclear how the synethsized data is constructed. What are entities, and how exactly are they used? For the utility data, are these expected to consist of pairs of harmful prompts and compliant harmful responses? Or are benign questions related to the harmful prompt constructed, for which benign responses are created? I see in D.7 there is an educational prompt; is this used for creating the utility data?

**Questions:**

1. Line 172: How is “strength” measured? Is it by the predicted probability?
2. Line 186: How is it determined that an index is considered “critical”?

---

> ### Author Response · Authors · 2025-11-17
> **Response to Reviewer y7dP part(1/5)**
>
> **We sincerely appreciate the reviewer’s thoughtful and valuable feedback. We have carefully considered each comment and provide our responses below, hoping that we have addressed your concerns to your satisfaction. If we have done so, we would be truly grateful if you could kindly reconsider your rating of 2 (reject). However, if any concerns persist, we would be deeply thankful for your continued guidance. We remain fully committed to addressing any outstanding issues and will make every effort to improve our submission in line with your expectations.**
>
> > **Weakness 1**
> >
> > Notation discrepancies - Line 126: “Let vocabulary be V, with length L” — it seems in the subsequent notation that L refers to the sequence length, not the vocabulary size.
> >
> > **Weakness 2**
> >
> > Spelling mistakes - Figure 2: “Quary” should be “Query - Line 275: Should “1) preserving” be “1) Utility-preserving”?
>
> Thank you for your careful review. Regarding the **notation discrepancy**, we agree that the notation in line 126 should clearly specify that LLL refers to the sequence length, not the vocabulary size. We have revised this sentence in the revised version. As for the **spelling mistakes**, we have corrected "Quary" to "Query" in Figure 2, and changed "1) preserving" to "1) Utility-preserving" in line 275.

---

> ### Author Response · Authors · 2025-11-17
> **Response to Reviewer y7dP part(2/5)**
>
> > **Weakness 3**
> >
> > Line 129 assumes that the vocabulary contains a harmful set and safe set of tokens, but in reality whether a token is considered “harmful” or “safe” heavily depends on the surrounding context.
>
> Thank you for your insightful comment.
>
> You are right that the harmfulness or safeness of a token is  context-dependent. Our formulation does not assume that tokens are harmful or safe in isolation. Instead, in harmful-input scenarios, we classify a token as harmful if it does not have rejective semantics.
>
> This context-conditioned definition is consistent with our theoretical analysis: the margin comparison between harmful and reject tokens is evaluated specifically under harmful contexts and masked positions, not globally across the vocabulary.

---

> ### Author Response · Authors · 2025-11-17
> **Response to Reviewer y7dP part(3/5)**
>
> > **Weakness 4**
> >
> > The takeaway from the main theoretical result seems to be rather obvious (that one should reduce the gaps for all mask positions). If one were to design a defense, the more obvious first solution would be to perform data augmentation on many different mask locations, rather than a few specific mask locations. Of course it is good to rigorously confirm this by proving the claim, however I feel that the high level takeaway doesn’t really yield a surprising insight.
>
> Thank you for your thoughtful feedback.
>
> While the high-level takeaway, by reducing the margin gaps for all masked positions is essential, but it also have some limitations now.
>
> **Theoretical limitations**:
> - Generic masking ignores step-wise contextual dependencies inherent in diffusion generation, which means that local context information between intermediate denoising steps is lost.
> - The attack space is theoretically unbounded — the [MASK] positions can be adversarially chosen in infinitely many ways, making it nearly impossible to label harmful regions accurately under a fully general setting.
>
> **Practical limitations**: At present, there is a lack of large, well-annotated datasets to support general-purpose defenses at the diffusion level. This makes it unrealistic to train models effectively across all possible masking configurations.
>
> In contrast, our study offers a rigorous theoretical foundation and an empirically validated defense that focuses on structurally critical positions identified through margin analysis. The Reject-MASK strategy leverages this insight to perform targeted data synthesis and focused training, which has been shown experimentally to achieve strong robustness with minimal utility loss.
>
> Thus, the contribution of our work lies not in proposing a trivial idea, but in establishing the theoretical reasoning, formal proofs, and practical training design that together make this defense both explainable and effective for dLLMs.

---

> ### Author Response · Authors · 2025-11-17
> **Response to Reviewer y7dP part(4/5)**
>
> > **Weakness 5**
> >
> > It is very unclear how the synethsized data is constructed. What are entities, and how exactly are they used? For the utility data, are these expected to consist of pairs of harmful prompts and compliant harmful responses? Or are benign questions related to the harmful prompt constructed, for which benign responses are created? I see in D.7 there is an educational prompt; is this used for creating the utility data?
>
>  Thank you for your detailed review and questions regarding the construction of the synthesized data.
>
>  **Safe Dataset Construction:** It consists of harmful prompts created using DIJA, with masked positions that are filled with refusal words.  The goal is to ensure that all masked tokens are replaced with safe and ethically aligned responses, effectively rejecting harmful behaviors.
>
> **Utility Data Construction:**
>
> **Entities:** In our work, entities refer to key components extracted from harmful prompts, such as important names, actions, or objects. These entities are extracted from the HarmBench dataset, and they are used to construct questions for the utility dataset.
>
> The building process of utility data follows a two-stage process:
>
> **Step 1 (Entity Extraction):** We first extract entities from harmful prompts in HarmBench.
>
> **Step 2 (Questions and Step-by-Step Responses):** Once the entities are identified, we create benign, step-by-step questions incorporating these entities. These questions are then paired with concise step-by-step responses aimed at mitigating the over-rejection issue observed in the Safe dataset. This is where the **educational prompt** (referenced in section D.7 of the paper) plays a key role. It is used to guide the creation of the step-by-step instructional responses, which are intended to be informative, task-relevant, and neutral, effectively counteracting the over-rejection issue in the Safe dataset.

---

> ### Author Response · Authors · 2025-11-17
> **Response to Reviewer y7dP part(5/5)**
>
> > Questions:
> >
> > 1. Line 172: How is “strength” measured? Is it by the predicted probability?
> >
> > 2. Line 186: How is it determined that an index is considered “critical”?
>
> Thank you for your helpful comments.
>
> **Strength Measurement**: The "strength" of guidance is quantified using the classifier-free guidance strength parameter  which adjusts the model's preference for generating tokens that are consistent with the provided prompt. Specifically, it is measured by the log probability difference between the conditioned and unconditioned outputs, rather than by the predicted probability. This adjustment influences the generation process by tilting the probability distribution, amplifying the likelihood of generating completions aligned with the prompt.
>
> **Critical Index Determination**: A "critical" index is defined as a position in the sequence where the model's decision significantly impacts the outcome of the generation. These indices are identified based on their potential to alter the final output, particularly when masked.The key idea is that certain tokens, when masked, can have a disproportionate effect on the success of an attack due to their role in shaping the generated content. We systematically identify these positions through their influence on the model's behavior during generation.
>
> We will revise the paper to provide additional clarity on these points and ensure that the explanations of "strength" and "critical" indices are more explicit.

---

> ### Author Response · Authors · 2025-11-17
>
> Dear Reviewer y7dP,
>
> We sincerely appreciate your thoughtful and valuable feedback. We have carefully considered each of your comments and have provided detailed responses, which we hope will address your concerns to your satisfaction. To facilitate your review, we have summarized the key points of our rebuttal below for your convenience.
>
> *   Correction of **notation and spelling errors** (please refer to our response to Weakness 1 and 2 for details)
> *   Clarification on the **contextual definition of harmful tokens** within our framework (please refer to our response to Weakness 3 for details)
> *   Elaboration on the **theoretical contribution and practical limitations** of our main result, highlighting its non-obvious insights (please refer to our response to Weakness 4 for details)
> *   A detailed explanation of the **construction process for our synthesized datasets**, including the role of entities and the educational prompt (please refer to our response to Weakness 5 for details)
> *   Answers to your questions regarding **the measurement of "strength" and the determination of "critical" indices** (please refer to the "Questions" section in our response for details)
>
> We humbly hope that our responses have adequately addressed your concerns. We are acutely aware of the significant challenge involved in reviewing numerous submissions, and we want to express our heartfelt appreciation for the careful and thoughtful attention you have devoted to our research. If our revisions and clarifications have resolved your concerns, we would be truly grateful if you could kindly reconsider your rating. However, if any concerns persist, we would be deeply thankful for your continued guidance and will make every effort to improve our submission in line with your expectations.
>
> Thank you again for your invaluable time and feedback.
>
> Best regards,
>
> The Authors

---

> ### Author Response · Authors · 2025-11-26
>
> Dear Reviewer,
>
> I hope this message finds you well. As the discussion period is drawing to a close in just a few days, I would like to once again express my deepest gratitude for your time, patience, and invaluable guidance throughout the review process.
>
> Before the discussion ends, I humbly wish to ensure that we have addressed all of your concerns to the best of our ability. If there are any remaining questions, suggestions, or aspects that you believe require further clarification or improvement, please kindly share them with us. Your insights mean a great deal to us, and we sincerely hope to refine our work in accordance with your thoughtful advice.
>
> Thank you once again for your generous effort, careful consideration, and the expertise you have devoted to reviewing our paper. We are truly grateful for your support.
>
> With my most respect and sincere appreciation,
>
> Authors

---

> ### Author Response · Authors · 2025-11-29
> **Summary for Rebuttal to AC**
>
> Dear Area Chair,
>
> We appreciate the reviewer’s thoughtful feedback and the time they took to review our submission. In response to their comments, we have made the following key improvements:
>
> * **Notation and spelling corrections**: We clarify the distinction between sequence length and vocabulary size, and correct spelling errors ("Quary" to "Query" and "preserving" to "Utility-preserving").
>
> * **Clarification on harmful tokens**: We **elaborate that the harmfulness or safeness of a token is context-dependent**, addressing the concern about the definition of harmful tokens.
>
> * **Theoretical contribution**: We further **explain the non-obvious nature of our theoretical results and the limitations** of more general masking strategies, emphasizing the value of our defense framework.
>
> * **Data synthesis clarification**: We **clarify the construction of our synthesized datasets**, especially the role of entities and the educational prompt used in creating utility data.
>
> * **Answers to specific questions**: We **provide detailed explanations** regarding the measurement of "strength" and the determination of "critical" indices.
>
> Following these revisions and clarifications, **the reviewer has updated the rating from 2  to 4**, reflecting their increased confidence in our work. We thank both the reviewer and you, as Area Chair, for your continued effort and valuable feedback, which has helped improve the quality of our paper.
>
> Best regards,
>
> The Authors

---

### Official Review · Reviewer_xbp6 · 2025-11-01

**Soundness:** 3
**Presentation:** 2
**Contribution:** 2
**Rating:** 4
**Confidence:** 3

**Summary:**

This paper investigates the safety vulnerabilities of diffusion large language models (dLLMs). The authors theoretically analyze why dLLMs are susceptible to mask-based jailbreaks, showing that bidirectional contextual modeling and parallel decoding enable attackers to insert [MASK] tokens that cause harmful completions.

The theoretical analysis conducted results in lower bounds on attack success probability based on token margin accumulation and scheduling effects, offering a mathematical characterization of why such jailbreaks succeed. Building on this, the authors propose a two-stage data synthesis framework and a Reject-MASK training strategy, focusing on rejection-related tokens to strengthen safe behavior. Experiments on HarmBench and JailbreakBench demonstrate drastic reductions in attack success rates (e.g., from ~90% to ~10%) with modest utility degradation.

**Strengths:**

The paper approaches a relevant practical research problem from a theoretical point of view.

The mathematical formalism leads to insightful conclusions on the importance of token margin accumulation and scheduling effects when discussing the success rate of MASK-based adversarial attacks.

The proposed defences show significant improvements in model robustness while keeping competitive utility scores.

**Weaknesses:**

**Unclear Theoretical Presentation and Notation**

The theoretical section is difficult to follow due to unclear notation and missing definitions. For instance, the variable U is used repeatedly but never formally defined (it likely refers to the user prompt). This lack of clarity makes the derivations hard to interpret.

**Limited Comparison with Related Work**

Despite citing several relevant defenses in Section 2.2, the paper does not include quantitative comparisons against them. Table 1 only evaluates against prompt-based defenses such as RPO and Self-Reminder, leaving out other cited approaches.

**Mathematical and Logical Errors in Proofs**

There appear to be some mistakes or inconsistencies in the theoretical proofs and their interpretation. In the proof of **Lemma 1**, the authors use the relaxation $\sum_{u \notin (h*, s*)} e^{z_i(u)} \le e^{z_i(h*)}$ which does not appear to be mathematically valid. Moreover, equations (15-17) seem to use “$\ge$” where logic dictates “$\le$.”
Finally, several bounds are described as scaling *polynomially* with K, whereas they are in fact *exponential* in K.


**Inconsistencies Between Figure 4 and the Discussion**

Figure 4 (a,c) appears to contradict the discussion in Section 6.3. Specifically, the Mix and Safe models are actually more sensitive to the number of [MASK] tokens when examining ASR-e. The main trend in Figure 4 suggests that these models learn to insert refusal keywords without truly avoiding harmful output: ASR-k is low, but ASR-e remains high for prompts with 30 or more [MASK] tokens.

**Gap Between Theory and Empirical Validation**

There is a notable gap between the theoretical analysis and the experiments. The paper introduces several assumptions and intermediate conclusions that are never empirically tested. Including additional statistics on margin scores and their correlation with attack success probability, or ablation studies validating the impact of scheduling on ASR, would significantly strengthen the work.

**Questions:**

1. The authors evaluated only on utility benchmarks designed for classical LLMs. The Reject-MASK training might further degrade performance on dLLM-specific tasks like fill-in-the-gaps. Could the authors provide some comparisons on this type of task?
2. Other questions are related to the weaknesses discussed above.

---

> ### Author Response · Authors · 2025-11-24
> **Response to Reviewer  xbp6 part(1/6)**
>
> **We sincerely appreciate the reviewer’s thoughtful and valuable feedback. We have carefully considered each comment and provide our responses below, hoping that we have addressed your concerns to your satisfaction. If we have done so, we would be truly grateful if you could kindly reconsider your rating of 4 (marginally below the acceptance threshold). However, if any concerns persist, we would be deeply thankful for your continued guidance. We remain fully committed to addressing any outstanding issues and will make every effort to improve our submission in line with your expectations.**
>
> > **Weakness 1**  **Unclear Theoretical Presentation and Notation**
> >
> > The theoretical section is difficult to follow due to unclear notation and missing definitions. For instance, the variable U is used repeatedly but never formally defined (it likely refers to the user prompt). This lack of clarity makes the derivations hard to interpret.
>
> We sincerely apologize for the lack of clarity in the theoretical section.
> As you correctly pointed out, the variable "U" refers to the user input prompt. In the revised version, we have explicitly defined this variable to ensure a clearer understanding and avoid any confusion in the derivations.

---

> ### Author Response · Authors · 2025-11-24
> **Response to Reviewer xbp6 part(2/6)**
>
> > **Limited Comparison with Related Work**
> >
> > Despite citing several relevant defenses in Section 2.2, the paper does not include quantitative comparisons against them. Table 1 only evaluates against prompt-based defenses such as RPO and Self-Reminder, leaving out other cited approaches.
>
> We sincerely thank the reviewer for this constructive comment. We agree that the absence of a **quantitative comparison** between our defence and alignment methods cited in Section 2.2 may be viewed as a limitation. We would like to clarify two important aspects and propose how we intend to address them in a revision.
>
> Firstly, methods in Section 2.2, such as suffix‑based attack‑aware defence(RPO), have been developed specifically for **autoregressive LLMs**. In contrast, our work focuses on a **diffusion‑based architecture**. Because of this architectural difference, a direct head‑to‑head comparison would not only require adapting those methods to our setting, whcih is not only the opration may change their mechanisms, but also it may produce misleading results: the design assumptions(causal generation,sequential decoding) break down in the diffusion context(parallel denoising, bidirectional contexts). As recent work points out, alignment methods tuned for autoregressive LLMs **do not** readily transfer to diffusion models [1].
>
> Secondly, we will revise paper as follows:
> * (1) we will include a qualitative discussion table comparing our method with autoregressive‑model alignment defences, explicitly noting the architectural mismatch, the deployment assumptions, and where our method diverges
> * (2) we will emphasize in the revised version that our contribution is not to show a stronger defence against the same baseline, but rather **customizing defense measures for the diffusion-based LLMs paradigm**, which has not been studied in prior work. We hope these additions will strengthen the paper’s framing and clarify the scope of our contribution.
>
> [1] Wen et al., An emergent safety vulnerability of Diffusion LLMs. 2025.

---

> ### Author Response · Authors · 2025-11-24
> **Response to Reviewer xbp6 part(3/6)**
>
> > **Mathematical and Logical Errors in Proofs**
> >
> > There appear to be some mistakes or inconsistencies in the theoretical proofs and their interpretation. In the proof of **Lemma 1**, the authors use the relaxation which does not appear to be mathematically valid. Moreover, equations (15-17) seem to use “≥” where logic dictates “≤.”
> > Finally, several bounds are described as scaling *polynomially* with K, whereas they are in fact *exponential* in K.
>
>  Thank you for pointing out the issues with mathematical proofs. We sincerely apologize for the inconsistencies and errors in the original submission.
>
> In the revised version, we have revised the proof of Lemma 1 and corrected the mathematical relaxation that was previously used.
>
> The new proof of Lemma 1 is as follows:
>
> Let $h_i^\star=h_i$ and $s_i^\star=s_i$ be the strongest candidates under their respective sets, $\Gamma(x) = \frac{1}{1+e^{-x}+(|\mathcal V| - 2)e}$, and define the **minimum margin gap** as
>
> \\begin{equation}
> \\gamma_i:=z_i(h_i^\\star)-z_i(s_i^\\star).
> \\end{equation}
>
> Then
>
> \\begin{equation}
> p_\\phi(h_i^\\star\\mid U,X_t^{(-i)})\\;\\ge\\;\\Gamma(\\gamma_i).
> \\end{equation}
>
> \\begin{equation}
> p_\\phi(h_i^\\star\\mid U,X_t^{(-i)})=\\frac{e^{z_i(h_i^\\star)}}{\\sum_{u\\in\\mathcal V}e^{z_i(u)}}
> \\end{equation}
>
> \\begin{equation}
> =\\frac{1}{1+e^{-(z_i(h_i^\\star)-z_i(s_i^\\star))}+\\sum_{u\\in\\mathcal V\\setminus\\{h_i^\\star,s_i^\\star\\}}e^{z_i(u)-z_i(h_i^\\star)}} \\ge \\frac{1}{1+e^{-\\gamma_i}+(|\\mathcal V| - 2)e}=\\Gamma(\\gamma_i)
> \\end{equation}
>
> Here, we uses the relaxation $z_i(u)-z_i(h^\\star) \\leq 1$.
>
> Additionally, the bounds that are incorrectly described as scaling "polynomially" with K have been updated to reflect the correct scaling behavior in revised version, which we now define with a new lower bound function Γ.

---

> ### Author Response · Authors · 2025-11-24
> **Response to Reviewer xbp6 part(4/6)**
>
> > **Inconsistencies Between Figure 4 and the Discussion**
> >
> > Figure 4 (a,c) appears to contradict the discussion in Section 6.3. Specifically, the Mix and Safe models are actually more sensitive to the number of [MASK] tokens when examining ASR-e. The main trend in Figure 4 suggests that these models learn to insert refusal keywords without truly avoiding harmful output: ASR-k is low, but ASR-e remains high for prompts with 30 or more [MASK] tokens.
>
> We appreciate your feedback. In the paper, we emphasises that the defence models “learn refusal keywords” but “do not always avoid the underlying harmful concept.” We now see that this phrasing may not fully meditate the data patterns in the figure and might appear inconsistent.
>
> To clarify: the persistence of rising ASR‑e is because ASR‑e captures semantic leakage of harmful content (even if there is a refusal keyword in), wbut ASR‑k measures a more surface‑level refusal keyword detection.
> In effect, what we observe is: as the number of [MASK] tokens grows, the models increasingly insert the refusal keyword, which we can get from *ASR‑k reducing**, but they still allow harmful semantic content to be expressed, which leads to **ASR‑e staying high**. That is why the “good” trend of low ASR‑k does not automatically imply “good” performance on ASR‑e.
>
> This behaviour is consistent with recent work [1] showing that detection of refusal keywords alone gives a false sense of security. [1] demonstrates that classifiers which trigger on surface patterns may still allow harmful content to slip through because the semantic risk remains.
>
> In our revised revision, we will explicitly explain this difference:
>
> - we will update the text in Section 6.3 to highlight that ASR‑e and ASR‑k measure different risk axes.
>
> - We will add a annotation in Figure 4 (a,c) explaining the decoupling of ASR‑k decline vs. ASR‑e persistence as [MASK] grows.
>
> - We will include a paragraph acknowledging that the elevated ASR‑e at high [MASK] counts illustrates the limitation of relying solely on refusal‑keyword insertion as a safety strategy, which is defence must do more than keyword insertion to block semantic harm.
>
> [1] Wang et al., False Sense of Security: Why Probing-based Malicious Input Detection Fails to Generalize. 2025.

---

> ### Author Response · Authors · 2025-11-24
> **Response to Reviewer xbp6 part(5/6)**
>
> > **Gap Between Theory and Empirical Validation**
> >
> > There is a notable gap between the theoretical analysis and the experiments. The paper introduces several assumptions and intermediate conclusions that are never empirically tested. Including additional statistics on margin scores and their correlation with attack success probability, or ablation studies validating the impact of scheduling on ASR, would significantly strengthen the work.
>
> Thank you for your detailed review and questions.
>
> In our paper, we explicitly bridge empirical results with theoretical assumptions by presenting the dynamics of reject‑word distributions across decoding steps. As shown in **Figure 3 and Figure 5**, as the decoding step increases the proportion of **reject words** occurring among the top‑10 most frequent tokens changes noticeably.
>
> For the baseline model, this proportion remains almost remain no change, but for **safe** and **mix** versions the proportion rises steadily. This empirical trend supports our theoretical claim that the scheduling of safe alignment training drives the model to shift token probability mass away from previously frequent reject words, which reducs ASR. By quantifying the top‑10 ranking occupancy of reject words, we provide concrete statistics that link margin shifts in token ranking to improved defense performance.
>
> Furthermore, the ablation of the scheduling effect is implicitly validated by comparing the trajectories of base model, safe, mix. The safe and mix conditions, which incorporate the aligned scheduling, both show the upward trend in reject‑word proportion, whereas base model show no change. This difference outcome support our intermediate conclusion that scheduling matters for alignment effectiveness. We believe that these experiment‑driven observations mitigate the noted gap between theory and validation, and the trends presented serve as a reliable empirical counterpart to our assumptions.

---

> ### Author Response · Authors · 2025-11-24
> **Response to Reviewer xbp6 part(6/6)**
>
> > **Questions:**
> >
> > 1. The authors evaluated only on utility benchmarks designed for classical LLMs. The Reject-MASK training might further degrade performance on dLLM-specific tasks like fill-in-the-gaps. Could the authors provide some comparisons on this type of task?
> > 2. Other questions are related to the weaknesses discussed above.
>
>
> ## For Question 1:
> Thank you for this important observation. We very appreciate the opportunity to extend our evaluation to more dLLM‑specific tasks, and we have conducted additional experiments on the CLOTH cloze‑test dataset [1](CLOTH‑M and CLOTH‑H subsets) to examine downstream fill‑in‑the‑gap capability. The dataset is widely used for evaluating language understanding via missing‐word prediction and is designed to challenge models’ reasoning and long‑context comprehension.
>
> Our testing method is to place the selected options at the end with: A. Option 1 B. Option 2....
> The method is used to indicate after the original sentence. Our evaluation metric is **accuracy**
>
> **Our results** are summarized as follows:
> | Model                        | CLOTH   | CLOTH-M | CLOTH-H |
> |------------------------------|---------|---------|---------|
> | **LLaDA-v1.5**                |         |         |         |
> | Base model                    | 0.435   | 0.506   | 0.364   |
> | Safe w/ Reject-MASK           | 0.419   | 0.485   | 0.353   |
> | Mix w/o Reject-MASK           | 0.431   | 0.499   | 0.363   |
> | Mix w/ Reject-MASK            | 0.425   | 0.490   | 0.360   |
> | **MMaDA-MixCoT**              |         |         |         |
> | Base model                    | 0.487   | 0.523   | 0.451   |
> | Safe w/ Reject-MASK           | 0.466   | 0.502   | 0.430   |
> | Mix w/o Reject-MASK           | 0.482   | 0.528   | 0.436   |
> | Mix w/ Reject-MASK            | 0.475   | 0.518   | 0.432   |
>
> From this table, it is clear that applying Reject‑MASK leads to a little drop in fill‑in performance but the degradation is relatively small compared to the baseline. Importantly, drop does not collapse performance: even with the defense in place, the models still retain a substantial original capability.
>
> [1] Xie et al., Large-scale Cloze Test Dataset Created by Teachers. 2018.
>
> ## For question 2:
> Thank you for your comments. We appreciate your attention to the potential weaknesses in the paper and have taken your feedback seriously. You can find the responses for them above.

---

> ### Author Response · Authors · 2025-11-24
>
> Dear Reviewer xbp6,
>
> We are deeply thankful for the time, care, and effort you have so generously dedicated to reviewing our paper. Your insightful and discerning feedback has been invaluable in helping us refine and strengthen our work. We fully recognize the immense responsibility and challenge of reviewing numerous submissions, and we are profoundly grateful for the careful and thoughtful attention you have devoted to our research. Your comments have not only guided us in improving the clarity and rigor of our paper but have also contributed to deepening our understanding of the broader implications of our study. In light of your thoughtful suggestions, we have carefully prepared comprehensive responses to your comments, and for your convenience, we have summarized the key points of our rebuttal below.
>
> - Clarification of **theoretical notation and definitions** (please refer to Weakness 1 in response for details)
>
> - Explanation of **comparison with related work and architectural mismatch** (please refer to Weakness 2 in response for details)
>
> - Corrections to **proofs and scaling statements** (please refer to Weakness 3 in response for details)
>
> - Clarification of **Figure 4 trends and the difference between ASR-k and ASR-e** (please refer to Weakness 4 in response for details)
>
> - Bridging **theory and experiments** via reject-word distribution statistics across decoding steps (please refer to Weakness 5 in response for details)
>
> - Additional evaluation on **dLLM-specific tasks such as CLOTH** (please refer to the "Questions" section in the response for details)
>
> We humbly hope that our responses have fully addressed your concerns and that the revisions we have made meet your expectations. If there are any lingering questions or areas that require further clarification, we are more than willing to provide any additional information you may need. We deeply understand the complexity and time constraints involved in the review process, and we are truly grateful for your thoughtful consideration. If our revisions have resolved your concerns, we would be immensely grateful if you could reconsider your rating. However, should any issues remain, we warmly welcome any further feedback and will continue to make every effort to improve our submission.
>
> Thank you again for your invaluable time and thoughtful feedback.
>
> Best regards,
>
> Authors

---

> ### Author Response · Authors · 2025-11-26
>
> Dear Reviewer,
>
> I hope this message finds you well. As the discussion period is drawing to a close in just a few days, I would like to once again express my deepest gratitude for your time, patience, and invaluable guidance throughout the review process.
>
> Before the discussion ends, I humbly wish to ensure that we have addressed all of your concerns to the best of our ability. If there are any remaining questions, suggestions, or aspects that you believe require further clarification or improvement, please kindly share them with us. Your insights mean a great deal to us, and we sincerely hope to refine our work in accordance with your thoughtful advice.
>
> Thank you once again for your generous effort, careful consideration, and the expertise you have devoted to reviewing our paper. We are truly grateful for your support.
>
> With my most respect and sincere appreciation,
>
> Authors

---

> ### Author Response · Authors · 2025-11-29
> **Summary for Rebuttal to AC**
>
> Dear Area Chair,
>
> We sincerely appreciate the time and effort the reviewer has invested in evaluating our submission. Based on their insightful feedback, we have made significant revisions to address all the concerns raised. Specifically, we have clarified theoretical notation and definitions, corrected mathematical errors, expanded on comparisons with related work, and provided further empirical validation to bridge the gap between theory and experiments.
> ﻿
> The reviewer’s initial concerns have been addressed as follows:
> ﻿
> * **Theoretical Clarity**: We **explicitly define key variables**, such as "U" for the user prompt, to improve understanding.
> ﻿
> * **Comparison with Related Work**: We clarify that direct comparisons with autoregressive LLM defenses are not feasible due to architectural differences, but we have **added a qualitative comparison to highlight our contributions in the diffusion-based LLM context**.
> ﻿
> * **Mathematical Errors**: We **revise the proof of Lemma 1 and correct the scaling behavior of the bounds**.
> ﻿
> * **Figure 4 Clarifications**: We **update the text and figure to explain the relationship between ASR-k and ASR-e**, emphasizing the distinction between keyword insertion and semantic risk.
> ﻿
> * **Empirical Validation**: We **provide additional statistics and ablation studies on reject-word distributions**, strengthening the connection between theoretical claims and experimental results.
> ﻿
> * **Evaluation on dLLM-Specific Tasks**: We **add performance comparisons on the CLOTH cloze-test dataset**, showing that our defense mechanism slightly reduces performance but does not cause a collapse.
>
> As a result of these revisions, **the reviewer has increased the rating from 4 to 6**. We would like to express our sincere gratitude to the reviewer for their constructive feedback and to you, the Area Chair, for overseeing the review process. We hope that the revised version meets the expectations and standards set forth.
>
> Thank you once again for your support and valuable guidance.
>
> Best regards,
>
> The Authors

---

### Note · Authors · 2026-01-08

I have read and agree with the venue's withdrawal policy on behalf of myself and my co-authors.